# Predicting Visceral Pleural Invasion in Resected Lung Adenocarcinoma via Computed Tomography

**DOI:** 10.3390/cancers17091414

**Published:** 2025-04-23

**Authors:** Chia-Cheng Kao, Hu-Lin Christina Wang, Mong-Wei Lin, Tung-Ming Tsai, Hsao-Hsun Hsu, Hsien-Chi Liao, Jin-Shing Chen

**Affiliations:** 1Department of Surgery, National Taiwan University Hospital, Taipei 100225, Taiwan; az820703@gmail.com (C.-C.K.); mwlin@ntu.edu.tw (M.-W.L.); ntuhsu@gmail.com (H.-H.H.); chenjs@ntu.edu.tw (J.-S.C.); 2Division of Thoracic Surgery, Department of Surgery, National Taiwan University Hospital, Taipei 100225, Taiwan; eldorado0607@gmail.com; 3Division of Traumatology, Department of Surgery, Far Eastern Memorial Hospital, New Taipei City 22000, Taiwan; drcwang@gmail.com; 4Department of Surgical Oncology, National Taiwan University Cancer Center, Taipei 106037, Taiwan; 5Department of Traumatology, National Taiwan University Hospital, Taipei 100225, Taiwan; 6Graduate Institute of Clinical Medicine, College of Medicine, National Taiwan University, Taipei 100233, Taiwan

**Keywords:** curative lung resection, preoperative imaging, prognostic factors

## Abstract

This study evaluates the use of computed tomography (CT) imaging to predict visceral pleural invasion (VPI) in lung adenocarcinoma. A retrospective analysis of 643 patients who underwent curative lung tumor resection at National Taiwan University Hospital (2011–2015) compared preoperative CT characteristics with postoperative pathology. Key factors associated with VPI include tumor size, solid part size, ground glass opacity (GGO %), tumor shape, border type, distance from the visceral pleura (DFVP), and invasion site. Statistical analysis confirmed that VPI is an independent poor prognostic factor. Accurate preoperative CT assessment helps surgeons determine the extent of resection and identify high-risk patients who may benefit from early systemic treatment. Incorporating CT imaging characteristics into clinical decision-making enhances diagnostic accuracy, optimizes treatment strategies, and improves patient outcomes. This study highlights CT imaging as a valuable non-invasive tool for predicting pleural invasion and guiding precise surgical management in lung adenocarcinoma patients.

## 1. Introduction

In patients with surgically resected lung adenocarcinoma, preoperative computed tomography (CT) is essential to localize the lesion and make decisions regarding the resection margin [1,2]. Previous published studies, inclusive of Tung-Ming Tsai [3], H. Zhang [4], and Jian Zhou [5], had shown the effectiveness of CT-guided localization methods for the lung nodules before definite surgical resection. Additionally, CT imaging characteristics provide a gateway for chest surgeons to determine the final prognosis before operations [6]; in particular, visceral pleural invasion (VPI) may provide additional information [7,8,9]. In Taiwan, with the prevalence of low-dose chest CT scans, the early detection rate of lung cancer has been increasing. According to the published research of Shun-Mao Yang [10] and Chih-Yu Chen [11], low-dose CT is effective at detecting early lung cancers and has a particularly high cancer detection rate. Thus, the issue of detecting visceral pleural invasion (VPI) is much more important before the malignancy treatment. In addition, previous published studies by the lung cancer research group of National Taiwan University Hospital (NTUH), including Mong-Wei Lin [12] and Pei-Hsing Chen [13], had proved that one of the advanced patterns of early lung adenocarcinoma, tumor spread through air spaces (STAS), could be predicted as a prognostic factor by CT characteristics. As a result, in this study, we would like to figure out the impact of visceral pleural invasion (VPI), another advanced pattern of early lung adenocarcinoma.

According to a previous study, VPI is a size-independent poor prognostic factor in patients with stage I non-small cell lung cancer (NSCLC) (Huang, Wang [14]). Similar studies, including one by Seok and Lee [15], reported that VPI was a significant predictor of poor prognosis for small-sized (≤30 mm) partly solid lung adenocarcinoma. Iizuka and Kawase [16] created a prognostic algorithm assuming that VPI could be predicted using a score calculated from several simple CT findings and clinicopathologic factors. Nevertheless, pleural invasion in NSCLC is considered an indicator for upstaging to T2 or T3 in the eighth TNM staging system, and patients with pleural invasion should therefore be indicated for lobectomy, rather than sublobar resection [17]. Patients who undergo sublobectomy exhibit slightly lower survival rates than those who undergo lobectomy [18]. Some studies also recommend lobectomy instead of segmentectomy in patients with VPI (Jeon, Kim [19]), and even the extent of lymph node dissection (LND) was observed to vary with VPI status.

According to a study by Wo et al. [18], the optimal extent of LND varies according to VPI status, with T1-sized/VPI tumors (stage IB) requiring more extensive LND than T1-sized/non-VPI tumors (stage IA). Advanced differentiation of VPI patterns may enable chest surgeons to make more accurate decisions regarding the surgical strategy; therefore, accurate preoperative prediction of pleural invasion is essential for surgical planning. The current TNM staging system for the pathological status of pleural invasion classifies the extent of pleural invasion into four classes: PL0 (no pleural involvement, designated as no pleural invasion), PL1 (tumor invasion of the elastic layer of the visceral pleura, without reaching the visceral pleural surface), PL2 (tumor invasion to the visceral pleural surface), and PL3 (tumor invasion of the parietal pleura or chest wall). The TNM staging system defines PL1 and PL2 as VPI (staged as T2) and PL3 as pleural invasion (staged as T3).

The extent of pleural invasion is an important parameter for TNM score upstaging and indicates different outcomes. A previous study revealed that patients with PL0 and PL1 have similar survival rates and are considered to have irregularly shaped tumors; conversely, PL2 staging affects survival outcomes [20]. Regarding the CT images, we can conduct a more detailed differentiation of pleural invasion using different imaging characteristics. Previous research by Jia-Jun Wu [21] had collected early lung cancer patients from two large-volume medical centers in Taiwan and made an in-depth exploration of the relationship between CT imaging and recurrence. Another similar published study by Tzu-Ning Kao [22] had also found CT-based radiomic characteristics helpful for preoperative prediction of tumor invasiveness in lung adenocarcinomas presenting as pure ground-glass nodules. For nodal metastasis, You-Wei Wang [23]’s study also found the relation between CT characteristics and pathologic results. Thus, noninvasively characterizing the extent of tumor invasion based on imaging has been an attractive topic for chest surgeons and radiologists. For example, Imai et al. reported that measuring the arch distance-to-maximum tumor diameter ratios may be a simple, noninvasive technique for evaluating pleural invasion using CT [24]; when the ratio was >0.9, the sensitivity and specificity for thoracic invasion were 89.7% and 96.0%, respectively.

Pleural tags may also be indicators that can predict VPI via CT imaging [25,26,27]. Some studies classify pleural tags into three types: type 1 (one or more linear pleural tags), type 2 (one or more linear pleural tags with a soft tissue component at the pleural end), and type 3 (one or more soft tissue, cord-like pleural tags) [28]. In this study, no pleural invasion was found in the absence of pleural tags; conversely, observing type 2 pleural tags on conventional CT images can increase the accuracy of early VPI diagnosis. Nevertheless, using the CT characteristics of VPI cannot completely replace the current clinical staging for lung cancer. Another study mentioned that the CT features of VPI were not independent prognostic factors for disease-free survival in patients with clinical T1 lung adenocarcinoma [29], and although this conclusion argues against the use of the CT features of VPI as T2 descriptors in the clinical staging of lung cancer, they can still be used to make deductions.

Over time, more advanced imaging studies have been conducted; however, in areas concerning the preoperative prediction of VPI via imaging studies, many unknowns remain. Thus, the aim of this study was to evaluate the association between CT characteristics and VPI in patients with surgically resected lung adenocarcinoma.

## 2. Materials and Methods

### 2.1. Study Population

This retrospective, single-center study included 1648 consecutive patients diagnosed with lung cancer who underwent curative-intent pulmonary resection at NTUH between January 2011 and December 2015. Patients were excluded if they lacked complete preoperative chest CT imaging, serum carcinoembryonic antigen (CEA) level data, or pulmonary function test results (Figure 1). Available clinicopathological profiles were retrospectively reviewed, and the correlations between VPI and CT characteristics were analyzed. Informed consent was obtained from all 643 patients before study entry, and data regarding their basic profile, preoperative CT imaging characteristics, and postoperative events were collected. The study was approved by the Human Research Ethics Committee of NTUH (202408169RINC).

### 2.2. Basic Profiles and Characteristics

The basic profiles of the 643 consecutive patients were reviewed in detail, and data including age, sex, body mass index, tobacco use, and presence of VPI were recorded; the presence of VPI was confirmed by a final pathology report. Tumor characteristics on the CT images were also reviewed and individually analyzed, then subsequently interpreted by experienced chest surgeons and radiologists. Characteristics included the distance from visceral pleura (DFVP), the tumor site (we classified the primary tumor site into six different groups: right upper lobe, right middle lobe, right lower lobe, left upper lobe, left lower lobe, and other), carcinoembryonic antigen (CEA) level, lung function (including forced expiratory volume in 1 s [FEV1] and forced vital capacity [FVC]), and operation methods (we classified this item into three different groups: wedge + LND, segmentectomy + LND, and lobectomy + LND). Interval changes followed up by CT before operation were also recorded. If the tumor size increased more than 2 mm, it was defined as an interval change 1. If the maximum diameter of consolidation/maximum tumor diameter (C/T ratio) of a tumor changed from ground glass opacity (GGO) to part-solid or solid, it was defined as an interval change 2. The C/T ratio was defined by consolidation/maximum tumor diameter and measured by one or two board-certified thoracic radiologists. Regarding comorbidities, if a patient had underlying diseases such as diabetes mellitus, hypertension, coronary artery disease, or chronic obstructive pulmonary disease, the comorbidity was labeled 1. If there was no underlying disease, the comorbidity was labeled 0.

Postoperative events—including pathological staging and comorbidities—were also reviewed, and the final pathological stage was confirmed by an experienced pathologist. The basic profiles of patients with and without VPI were generally similar and without statistical significance.

### 2.3. Potential Risk Factors for Pleural Invasion

Potential risk factors for pleural invasion that were visible on the CT images were all reviewed and analyzed; these included tumor size (cm), solid part size, pleural contact of arch distance (classified into three groups: ≤1 cm, 1–2 cm, and >2 cm), GGO (%) (composed of pure GGO, C/T ratio ≤ 50%, and C/T ratio > 50%), tumor shape (regular or irregular), border type (classified into four types: A, B, C, and D), DFVP (classified into three groups: 0 cm, ≤0.5 cm, >0.5 cm; see Figure 2 and Figure 3), depth, and invasion site (chest wall, mediastinal, or fissure). A *p*-value <0.05 was considered statistically significant.

### 2.4. Univariate and Multivariate Analyses of Prognostic Factors

For univariate analysis, we included potential prognostic factors for adenocarcinoma with/without pleural invasion in the analysis, including tumor size (cm), solid part size, GGO (%), C/T ratio (%), tumor shape (regular vs. irregular), border type, DFVP, and invasion site. We calculated the odds ratio of each variable, and *p*-values <0.05 were considered statistically significant. Additionally, we performed multivariate analysis of the potential prognostic factors mentioned above.

### 2.5. Statistical Analysis

All variables in this study were collected and analyzed using the Chi-squared test, and odds ratios and *p*-values were recorded; we also performed a multivariate analysis of the potential prognostic factors, and the results were recorded.

## 3. Results

### 3.1. Patient and Tumor Characteristics

Of the 643 patients, 87 (13.5%) and 556 (86.5%) patients were classified into the VPI and non-VPI groups, respectively. The mean age was 60.02 (30.00–88.00) years, and there were no significant differences between the two groups (VPI: 61.61 years; non-VPI: 59.78 years). The ratio of males to females between the two groups was approximately 1:2 (32.5% to 67.5%), and the mean body mass index was 23.55. The proportion of smokers was 13.1% (84 of 643 patients). Regarding tumor site, the right upper and lower lobes were the dominant areas, accounting for 30.6% and 29.7% of patients, respectively. This was followed by the left lower lobe, right middle lobe, left upper lobe, and other, accounting for 19.6%, 11.0%, 8.9%, and 0.2% of patients, respectively. The percentage between the two groups was without statistical significance (*p* = 0.925). A normal CEA level was observed in 607 patients (mean, 94.4%; 87.4% in the VPI group and 95.5% in the non-VPI group), which was statistically significant (*p* = 0.002). The mean FEV1 and FVC were 108.57 (VPI: 108.32; non-VPI: 108.61) and 111.36 (VPI: 106.86; non-VPI: 112.06), respectively; no statistical significance was observed for these items (Table 1).

In the VPI group, the main operation method was lobectomy + LND, including 66 patients (75.9%); this was followed by wedge resection + LND (18.4%) and segmentectomy + LND (5.7%). In the non-VPI group, the main operation method was wedge resection + LND, including 244 patients (43.9%), followed by lobectomy + LND (40.6%); segmentectomy + LND was only performed in 1.5% of patients. The *p*-value for this item was <0.05, indicating statistical significance (Table 1).

Statistical significance was also observed regarding the pathological stage; in the VPI group, most patients were stage IIA (69.0%), followed by stage IIIB (9.3%) and stage IB (9.2%). In contrast, stage IB was dominant in the non-VPI group (90.8%), followed by stage IIIB (2.7%) and stage IIA (2.5%). Lastly, there was no statistical significance regarding the comorbidity rate between the two groups, and the mean comorbidity percentage was 49.8% (56.3% in the VPI group and 48.7% in the non-VPI group) (Table 1).

#### 3.1.1. Risk Factors for Pleural Invasion

Factors found to have statistical significance during analysis included tumor size, solid part size, GGO (%), C/T ratio (%), tumor shape, border type, DFVP, and invasion site. The mean tumor sizes in the VPI and non-VPI groups were 2.45 (0.66–4.73) cm and 1.67 (0.38–5.03) cm, respectively. Similarly, the mean solid part size in the non-VPI group was 0.63 (0.00–2.99) cm, smaller than that observed in the VPI group (1.67 [0.00–2.99] cm). GGO (%) was observed in 70.00% of patients in the VPI group and 47.61% of patients in the non-VPI group. Likewise, the C/T ratio (%) in the VPI group was 0.69, larger than that observed in the non-VPI group (0.30) (Table 2).

The proportion of irregular to regular tumor shapes was approximately 2:1 in the VPI group, which was statistically significant when compared with that in the non-VPI group (1:1). In the VPI group, the predominant border type was B (40.2%). In contrast, the main border type in the non-VPI group was A (49.3%). Nevertheless, the ratio of both border types C and D were larger in the VPI group than that in the non-VPI group (C, 25.3% to 12.6%; D, 25.3% to 11.7%). The mean DFVP was smaller in the VPI group (0.64 [0.00–4.24] cm) than it was in the non-VPI group (1.03 [0.0–5.91] cm); this was statistically significant. Finally, the invasion site was also significant. The main site in both groups was the chest wall; however, this only accounted for 66.7% of the VPI group, smaller than the percentage in the non-VPI group (78.2%). In the VPI group, the ratio of the tumor-invaded fissure was also larger than that in the non-VPI group (23.0% and 11.7%, respectively) (Table 2).

#### 3.1.2. Univariate Analyses of Prognostic Factors

Using univariate analyses, solid part size, GGO (%), C/T ratio (%), tumor shape, border type (comparing type A and type D), DFVP (cm), and invasion site (comparing chest wall and fissure) were found to have statistical significance (Table 3).

## 4. Discussion

Along with the improvement of CT in the 21st century, VPI is a valuable topic which has been frequently discussed among chest specialists and radiologists, particularly within the last decade. According to previous studies, VPI is known to be a size-independent poor prognostic factor in patients with stage I NSCLC [14], especially in those with small-sized (≤30 mm), partly solid lung adenocarcinomas [15]. Regarding NSCLC, pleural invasion is considered an indicator for upstaging to T2 or T3 in the eighth TNM staging system, which requires different treatment plans against the tumor. Thus, differentiating between the presence or absence of VPI via CT imaging characteristics before surgery would enable more precise decision-making by chest surgeons, thus allowing for a better prognosis.

In our study, we identified statistically significant potential risk factors for VPI; these included tumor size, solid part size, GGO (%), C/T ratio (%), tumor shape, border type, DFVP, and invasion site. A previous study created a score calculator (using scores calculated from several simple CT findings and clinicopathologic factors) to predict the presence of VPI [16]. The formula variants of CT characteristics included tumor diameter, tumor contact length with pleura, and tumor with cavity. We reached a similar consensus: tumor diameter may be a risk factor and predictor of VPI; however, our study revealed no statistical significance concerning the tumor contact length with the pleura, which may be due to different definitions. In the score calculator created by Iizuka et al., a tumor contact length with the pleura >16 mm was defined as a statistically significant predictor. In our study, the mean tumor contact lengths with the pleura were 5.3 mm and 6.4 mm in the VPI and non-VPI groups, respectively; these were much smaller than those defined in the previous study. Additionally, our study did not include tumors with cavities as a potential risk factor; instead, solid part size was used as a variant. In the VPI group, the solid part size was obviously larger than that in the non-VPI group and was statistically significant; this means that the probability of VPI increases with tumor progression.

GGO percentage was included in our study. As already known, GGO is defined as a hazy increase in lung attenuation that does not obscure the underlying bronchial or vascular structures. We found that the incidence of adenocarcinoma increased in correlation with an increasing ratio of solid components. Clinically, we often assume GGO to be a CT image characteristic with the potential for malignancy. In our study, the incidence of VPI corresponded with the GGO percentage; the higher the GGO percentage, the greater the likelihood that the final malignancy stage would be aggressive. Comparable results were observed regarding tumor shape. It is known that an irregular shape is a characteristic used to differentiate early lung cancer from a benign mass; thus, it may help to predict the presence of VPI.

In our study, border type was also a risk factor for VPI. Clinically, a pleural tag is often defined as one or more linear strands that extend from the nodule to the pleural surface due to thickening of the interlobular septa of the lung. Hsu et al. [28] classified pleural tags into three types: type 1 (one or more linear pleural tags), type 2 (one or more linear pleural tags with a soft tissue component at the pleural end), and type 3 (one or more soft tissue, cord-like pleural tags). No pleural invasion was found in the absence of pleural tags in this study. Additionally, observing type 2 pleural tags on conventional CT images can increase the accuracy of early VPI diagnosis. In the VPI group in our study, the most dominant border type was B, accounting for 40% of cases; this is consistent with the description of type 2 pleural tags in the study by Hsu et al. [28]. In the non-VPI group, the most dominant border type was A (the tumor was not attached to the pleura).

Last, our study found that DFVP was a risk factor. It is logical that the shorter the distance between the tumor and the pleura, the more likely the occurrence of VPI. The mean distances were 0.64 cm and 1.03 cm in the VPI and non-VPI groups, respectively. No previous study reported a cut-off point for the distance, and additional data may be required for calculations in the future.

### Limitations

Our study has some limitations. First, this was a retrospective study conducted at a single center and the study cohort was not initiated for this type of study. Furthermore, biases are inevitable, as different surgeons performed the operations, and the final CT image reports were interpreted by different radiologists. Second, statistical bias was hard to ignore; thus, we used different statistical methods to minimize bias. Third, our cohort included 643 cases collected between January 2011 and December 2015. The cohort size was medium and was not collected within the last five years; therefore, some collected data may have been lost. Furthermore, as the observation period concluded in 2015, more recent advancements in diagnostic modalities, including novel pleuroscopic techniques, were not available for analysis in this study [30,31]. However, due to the medium cohort size, the study will still provide statistical and clinical power to help us achieve precise decision-making and better prognosis. Last but not least, CT acquisition heterogeneity may introduce variability in the consistency and comparability of imaging data across different patients. To mitigate this potential bias, all CT scans in this study were performed using standardized imaging protocols, including uniform slice thickness, reconstruction algorithms, and contrast administration protocols.

## 5. Conclusions

VPI was a poor prognostic factor in early-stage NSCLC that could be predicted using CT imaging characteristics, including tumor size, solid part size, GGO (%), C/T ratio (%), tumor shape, border type, DFVP, and invasion site. The more accurately surgeons can predict the incidence of VPI, the more likely it is that they can achieve precise decision-making, potentially improving the patient’s outcome.

## Figures and Tables

**Figure 1 cancers-17-01414-f001:**
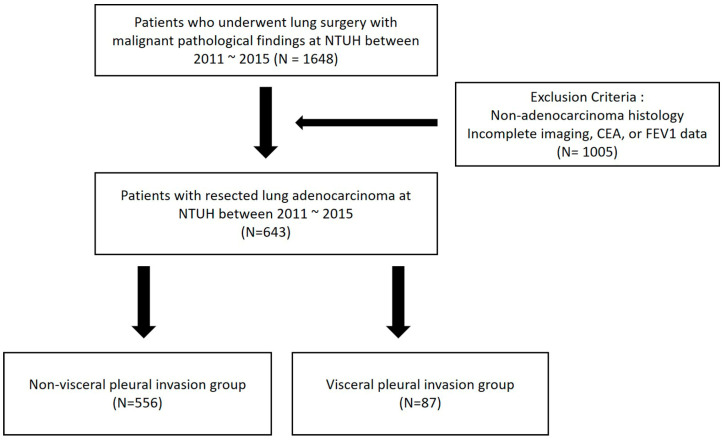
Flow diagram of patient enrollment and exclusion.

**Figure 2 cancers-17-01414-f002:**
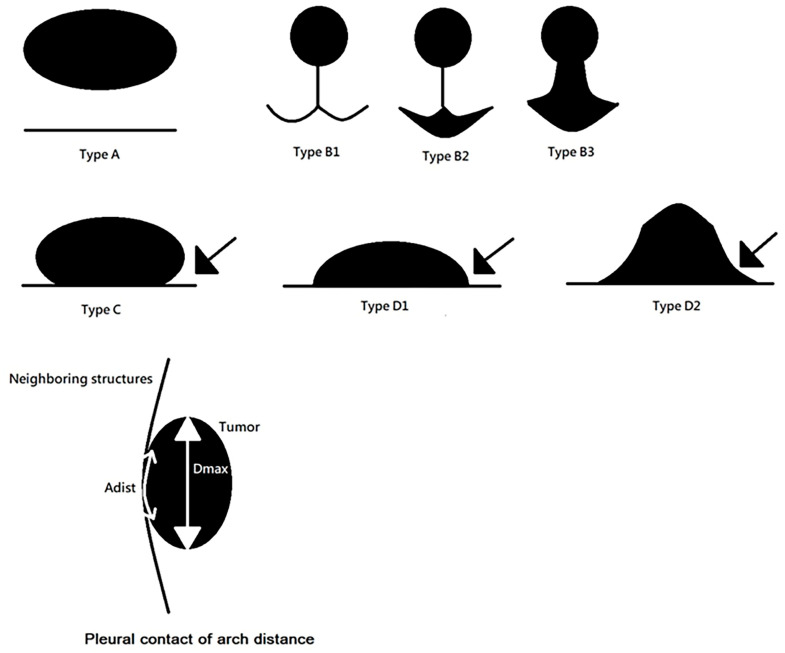
Distance from visceral pleura (DFVP) classification.

**Figure 3 cancers-17-01414-f003:**
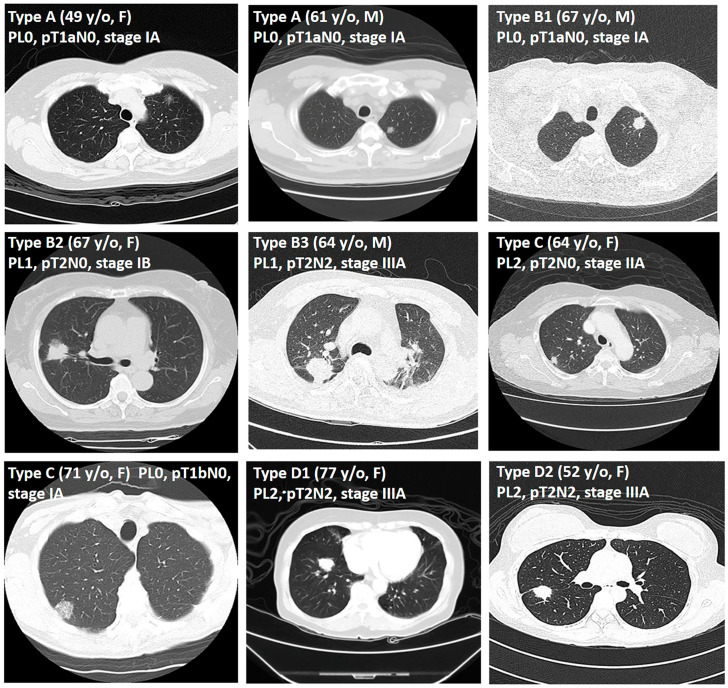
DFVP image examples. **Type A**, 49 y/o Female, left upper lobe 1.3 cm tumor, adenocarcinoma, PL0, pT1aN0, stage IA; 61 y/o Male, left upper lobe 1.9 cm tumor, adenocarcinoma, PL0, pT1aN0, stage IA. **Type B1**, 67 y/o Male, left upper lobe 1.9 cm tumor, adenocarcinoma, PL0, pT1aN0, stage IA. **Type B2**, 67 y/o Female, right upper lobe 2.5 cm tumor, adenocarcinoma, PL1, pT2N0, stage IB. **Type B3**, 64 y/o Male, right upper lobe 3.7 cm tumor, adenocarcinoma, PL1, pT2N2, stage IIIA. **Type C**, 71 y/o Female, right upper lobe 2.3 cm tumor, adenocarcinoma, PL0, pT1bN0, stage IA; 64 y/o Female, right upper lobe 1.8 cm tumor, adenocarcinoma, PL2, pT2N0, stage IIA. **Type D1**, 77 y/o Female, right lower lobe 2.5 cm tumor, adenocarcinoma, PL2, pT2N2, stage IIIA. **Type D2**, 52 y/o Female, right upper lobe 2.8 cm tumor, adenocarcinoma, PL2, pT2N2, stage IIIA.

**Table 1 cancers-17-01414-t001:** Patient and tumor characteristics.

	All Patients	Non-VPI	VPI	*p*-Value
**Case numbers**	643	556 (86.5%)	87 (13.5%)	
Age (years)	60.02 (30.00–88.00)	59.78 (31.00–88.00)	61.61 (30.00–87.00)	0.150
Sex				0.245
Male	209 (32.5%)	176 (31.7%)	33 (37.9%)	
Female	434 (67.5%)	380 (68.3%)	54 (62.1%)	
Body mass index (W/H^2^)	23.55 (13.24–38.94)	23.50 (13.24–38.94)	23.85 (16.86–34.28)	0.365
Smoking				367
Yes	84 (13.1%)	70 (12.6%)	14 (16.1%)	0.
No	559 (86.9%)	486 (87.4%)	73 (83.9%)	
Tumor Site				0.925
Right upper lobe	197 (30.6%)	171 (30.8%)	26 (29.9%)	
Right middle lobe	71 (11.0%)	63 (11.3%)	8 (9.2%)	
Right lower lobe	191 (29.7%)	163 (29.3%)	28 (32.2%)	
Left upper lobe	57 (8.9%)	51 (9.2%)	6 (6.9%)	
Left lower lobe	126 (19.6%)	107 (19.2%)	19 (21.8%)	
Other	1 (0.2%)	1 (0.2%)	0 (0.0%)	
CEA level				0.002 **
Normal	607 (94.4%)	531 (95.5%)	76 (87.4%)	
Abnormal	36 (5.6%)	25 (4.5%)	11 (12.6%)	
FEV1 (%)	108.57 (46.10–179.70)	108.61 (46.10–179.40)	108.32 (64.30–179.70)	0.895
FVC (%)	111.36 (46.10–179.70)	112.06 (46.10–179.40)	106.86 (64.20–143.80)	0.463
Operation method				0.000 ***
Wedge + LND	260 (40.4%)	244 (43.9%)	16 (18.4%)	
Segmentectomy + LND	91 (14.2%)	86 (1.5%)	5 (5.7%)	
Lobectomy + LND	292 (45.4%)	226 (40.6%)	66 (75.9%)	
Interval change				0.000 ***
0	156 (24.3%)	145 (26.1%)	11 (12.6%)	
1	157 (24.4%)	146 (26.3%)	11 (12.6%)	
2	330 (51.3%)	265 (47.7%)	65 (74.7%)	
Pathologic Stage				0.000 ***
IA	10 (1.6%)	10 (1.8%)	0 (0.0%)	
IB	513 (79.8%)	505 (90.8%)	8 (9.2%)	
IIA	74 (11.5%)	14 (2.5%)	60 (69.0%)	
IIB	15 (2.3%)	9 (1.6%)	6 (6.9%)	
IIIA	2 (0.3%)	0 (0.0%)	2 (2.3%)	
IIIB	24 (3.7%)	15 (2.7%)	9 (10.3%)	
IVA	4 (0.6%)	3 (0.5%)	1 (1.1%)	
IVB	1 (0.2%)	0 (0.0%)	1 (1.1%)	
Comorbidity				0.188
0	323 (50.2%)	285 (51.3%)	38 (43.7%)	
1	320 (49.8%)	271 (48.7%)	49 (56.3%)	

Abbreviations: CEA, carcinoembryonic antigen; FEV1, forced expiratory volume in 1 s; FVC, forced vital capacity; LND, lymph node dissection; VPI, visceral pleural invasion; W/H^2^, weight/height squared. If the tumor size increased more than 2 mm it was defined as interval change 1. If the C/T ratio of the tumor changed from GGO to part-solid or solid, it was defined as interval change 2. Regarding comorbidities, if the patient had underlying diseases, such as diabetes mellitus, hypertension, coronary artery disease, or chronic obstructive pulmonary disease, the comorbidity label was 1. If there were no underlying diseases, the comorbidity label was 0. ** *p* < 0.01, *** *p* < 0.001.

**Table 2 cancers-17-01414-t002:** Risk factors for pleural invasion.

		All Patients	Non-VPI (n = 556)	VPI (n = 87)	*p*-Value [Non–VPI vs. VPI]
**1**	Tumor size (cm)	1.78 (0.38–5.03)	1.67 (0.38–5.03)	2.45 (0.66–4.73)	0.000 ***
**2**	Solid part size	0.77 (0.00–2.99)	0.63 (0.00–2.99)	1.67 (0.00–2.99)	0.000 ***
**3**	PCAD (cm)	0.54 (0.00–3.60)	0.53 (0.00–3.60)	0.64 (0.00–2.63)	0.189
**4**	GGO (%)	50.64 (0.00–100.00)	47.61 (0.00–100.00)	70.00 (0.00–100.00)	0.000 ***
**5**	C/T ratio (%)	0.35 (0.00–1.00)	0.30 (0.00–1.00)	0.69 (0.00–1.00)	0.000 ***
**6**	Tumor shape				
	Regular	302 (47.0%)	275 (49.3%)	27 (31.0%)	0.001 **
	Irregular	341 (53.0%)	281 (50.5%)	60 (69.0%)	
**7**	Border type				0.000 ***
	A	282 (43.9%)	274 (49.3%)	8 (9.2%)	
	B	182 (28.3%)	147 (26.4%)	35 (40.2%)	
	C	92 (14.3%)	70 (12.6%)	22 (25.3%)	
	D	87 (13.5%)	65 (11.7%)	22 (25.3%)	
**8**	DFVP (cm)	0.97 (0.00–5.91)	1.03 (0.0–5.91)	0.34 (0.00–1.59)	0.000 ***
**9**	Invasion site				0.014 *
	Chest wall	493 (76.7%)	435 (78.2%)	58 (66.7%)	
	Mediastinal	65 (10.1%)	56 (10.1%)	9 (10.3%)	
	Fissure	85 (13.2%)	65 (11.7%)	20 (23.0%)	

Abbreviations: C/T, maximum diameter of consolidation/maximum tumor diameter; DFVP, distance from visceral pleura; PCAD, pleural contact of arch distance; GGO, ground glass opacity; VPI, visceral pleural invasion. * *p* < 0.05, ** *p* < 0.01, *** *p* < 0.001.

**Table 3 cancers-17-01414-t003:** Univariate and multivariate analyses of prognostic factors.

	Variable	Univariate Analysis		Multivariate Analysis		
		Odds Ratio	*p*-Value	Odds Ratio	95% CI	*p*-Value
**1**	Tumor size (cm)	1.028	0.713	1.063	0.776–1.455	0.705
**2**	Solid part size	3.719	0.000 ***	1.371	0.527–3.569	0.518
**3**	GGO (%)	1.017	0.000 ***	1.010	0.998–1.022	0.091
**4**	C/T ratio (%)	29.712	0.000 ***	5.928	0.551–63.842	0.142
**5**	Tumor shape (regular = 1, irregular = 0)	0.460	0.002 **	2.505	0.793–7.92	0.118
**6**	Border type (A = 1, D = 0)	0.086	0.000 ***	0.257	0.058–1.136	0.073
**7**	Border type (B = 1, D = 0)	0.703	0.257	0.337	0.091–1.253	0.105
**8**	Border type (C = 1, D = 0)	0.929	0.831	0.906	0.255–3.226	0.879
**9**	DFVP (cm) depth	0.647	0.002 **	0.610	0.381–0.977	0.04 *
**10**	Invasion site (chest wall = 1, fissure = 0)	0.433	0.004 **	0.903	0.245–3.319	0.877
**11**	Invasion site (mediastinal = 1, fissure = 0)	0.522	0.141	0.588	0.109–3.168	0.537

Abbreviations: CI, confidence interval; C/T, maximum diameter of consolidation/maximum tumor diameter; DFVP, distance from visceral pleura; GGO, ground glass opacity. * *p* < 0.05, ** *p* < 0.01, *** *p* < 0.001.

## Data Availability

The data presented in this study are available on request from the corresponding author due to privacy or ethical restrictions.

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
