# Peer review of "Predicting Visceral Pleural Invasion in Resected Lung Adenocarcinoma via Computed Tomography"

_cancers, 2025, doi:10.3390/cancers17091414_

Round 1
Reviewer 1 Report
Comments and Suggestions for Authors
The study describes the involvement of the visceral pleura or not in the staging of adenocarcinoma through CT images
Major criticisms
I believe that it is necessary to study the visceral pleura through thoracoscopy possibly associated with CT. CT cannot with its images replace trunoscopy for the study of the extension of metastasis from adenocarcinoma at the level of the visceral pleura.
The data reported are retrospective as indicated by the authors and refer to a very distant observation period that ends in 2015.
In the field of medicine there have been further procedures that should be analyzed in the diagnosis and accuracy of adenocarcinoma
Please see "Diagnostic utility of pleuroscopic guided pleural biopsy versus pleural fluid cell block in the diagnosis of malignant pleural effusion" by NG Bh et al 2025 where pleuroscopy is considered an essential method both for the qualitative study of pleural fluid and for biopsies allowing accurate differential diagnosis.
Author Response
Comments 1:
I believe that it is necessary to study the visceral pleura through thoracoscopy possibly associated with CT. CT cannot with its images replace trunoscopy for the study of the extension of metastasis from adenocarcinoma at the level of the visceral pleura.
Response 1:
Regarding the evaluation of visceral pleura involvement:
We fully agree with the Reviewer that computed tomography (CT) imaging alone cannot replace thoracoscopy for the detailed assessment of metastatic involvement of the visceral pleura. In our study design, CT imaging was utilized preoperatively to plan the thoracoscopic approach. Subsequently, thoracoscopy was performed intraoperatively, guided by the CT imaging findings, to confirm and delineate the extent of tumor invasion, which then informed the extent of surgical resection. Following the acquisition of the final pathological results, a correlation with the preoperative CT imaging findings is conducted.
Comments 2:
The data reported are retrospective as indicated by the authors and refer to a very distant observation period that ends in 2015.
In the field of medicine there have been further procedures that should be analyzed in the diagnosis and accuracy of adenocarcinoma
Response 2:
Regarding the retrospective nature and observation period:
We acknowledge the Reviewer’s comment regarding the retrospective nature of the study and the observation period ending in 2015. This was a limitation inherent to our study design. We have now added a limitation recognizing that more recent advances in diagnostic modalities, including novel pleuroscopic techniques, have emerged. Future prospective studies incorporating these newer technologies are certainly warranted to validate and expand upon our findings.
Comments 3:
Please see "Diagnostic utility of pleuroscopic guided pleural biopsy versus pleural fluid cell block in the diagnosis of malignant pleural effusion" by NG Bh et al 2025 where pleuroscopy is considered an essential method both for the qualitative study of pleural fluid and for biopsies allowing accurate differential diagnosis.
Response 3:
Regarding the reference to NG Bh et al. (2025):
Thank you for highlighting this important and recent publication. We have incorporated a citation of NG Bh et al. (2025) in the limitation of our revised manuscript (page 11, line 333-335).
We are grateful for the Reviewer’s insightful feedback, which has helped us improve the clarity and quality of our manuscript.
Reviewer 2 Report
Comments and Suggestions for Authors
Dear Editor,
The authors evaluated the relationship between CT features and visceral pleural invasion in patients with surgically resected lung adenocarcinoma. They retrospectively reviewed 643 patients with lung adenocarcinoma who underwent curative lung tumor resection between 2011 and 2015. In the study, VPI was found to be a poor prognostic factor in early-stage NSCLC that could be predicted using CT imaging features including tumor size, solid part size, GGO (%), C/T ratio (%), tumor shape, margin type, DFVP and invasion site. I congratulate the authors for this beautiful study that may benefit thoracic surgeons in predicting visceral pleural invasion.
Sincerely
Author Response
Comments 1:
Comments and Suggestions for Authors
Dear Editor,
The authors evaluated the relationship between CT features and visceral pleural invasion in patients with surgically resected lung adenocarcinoma. They retrospectively reviewed 643 patients with lung adenocarcinoma who underwent curative lung tumor resection between 2011 and 2015. In the study, VPI was found to be a poor prognostic factor in early-stage NSCLC that could be predicted using CT imaging features including tumor size, solid part size, GGO (%), C/T ratio (%), tumor shape, margin type, DFVP and invasion site. I congratulate the authors for this beautiful study that may benefit thoracic surgeons in predicting visceral pleural invasion.
Sincerely
Response 1:
Thank you very much for your positive and encouraging feedback. We greatly appreciate your thoughtful comments and are pleased that you found our study valuable for thoracic surgeons in predicting visceral pleural invasion. We have carefully reviewed the manuscript again and made minor edits to further improve the clarity and readability. We are grateful for your support and consideration of our work.
Reviewer 3 Report
Comments and Suggestions for Authors
Thank you for the opportunity to review the manuscript titled "Predicting Visceral Pleural Invasion in Resected Lung Adenocarcinoma via Computed Tomography." The authors retrospectively analyzed the CT features predictive of Visceral Pleural Invasion (VPI).
It would be beneficial if the authors could include a flow chart outlining the inclusion and exclusion criteria, as the exclusion criteria were not reported.
In the methods section, authors should also specify the CT machine used, the CT protocols employed, the expertise level of the thoracic radiologists who analyzed the images, and how measurements were taken for both solid and subsolid tumors.
Additionally, in the results section, authors should report the mean tumor distance that can predict pleural invasion.
The manuscript is of good quality, and I suggest some minor changes.
Author Response
Comments 1:
Comments and Suggestions for Authors
Thank you for the opportunity to review the manuscript titled "Predicting Visceral Pleural Invasion in Resected Lung Adenocarcinoma via Computed Tomography." The authors retrospectively analyzed the CT features predictive of Visceral Pleural Invasion (VPI).
It would be beneficial if the authors could include a flow chart outlining the inclusion and exclusion criteria, as the exclusion criteria were not reported.
Response 1:
Thank you for your valuable suggestion. In response, we have added a flowchart (Figure 1) to the revised manuscript to clearly illustrate the patient selection process, including both the inclusion and exclusion criteria. Additionally, we have included a detailed description of the exclusion criteria within the Methods section to enhance the transparency and reproducibility of our study design (page 3, line 131-135).
Figure 1. Flow diagram of patient enrollment and exclusion
Comments 2:
In the methods section, authors should also specify the CT machine used, the CT protocols employed, the expertise level of the thoracic radiologists who analyzed the images, and how measurements were taken for both solid and subsolid tumors.
Response 2:
We appreciate the Reviewer’s insightful comment. We have revised the Methods section to clarify that one or two board-certified thoracic radiologists with experience independently evaluated the CT images. Details on how measurements were performed for both solid and subsolid components have also been added. (page 4, line 149-151, line 161-163)
Comments 3:
Additionally, in the results section, authors should report the mean tumor distance that can predict pleural invasion.
Response 3:
Thank you for your valuable comment. In our study, we included a parameter termed Distance from Visceral Pleura (DFVP) to represent the tumor distance to the pleura. The mean DFVP was 0.64 [0.00–4.24] cm in the VPI group and 1.03 [0.0–5.91] cm in the Non-VPI group. We had concluded this information to the Results section (page 9, line 255-256). We appreciate your suggestion, which helped us clarify this aspect of our analysis.
Comments 4:
The manuscript is of good quality, and I suggest some minor changes.
Response 4:
Thank you very much for your encouraging feedback and constructive suggestions. We believe these revisions have further improved the quality and clarity of the manuscript.

Reviewer 4 Report
Comments and Suggestions for Authors
This paper investigated the association between CT imaging features and visceral pleural invasion in patients with surgically resected lung adenocarcinoma. It analyzes 643 cases using standard radiologic features and pathology-confirmed visceral pleural invasion.
This a potentially interesting paper. Please see some of the major concerns listed below:
Materials & Methods
Were all CT scans acquired with the same scanner and protocol across the 2011–2015 period? Were parameters like slice thickness, reconstruction algorithm, contrast administration, and radiation dose standardized?
How were features like “border type” or “tumor shape” operationally defined? Were they visually assessed or quantitatively measured using imaging software?
The dataset is over 10 years old. How do the authors justify the relevance of these findings to current imaging standards, which may now include dual-energy CT or radiomics?
Discussion:
The authors reference previous studies with similar findings but miss an opportunity to critically contrast their results with conflicting evidence, such as studies questioning the reliability of CT-based VPI prediction for early-stage tumors. Additionally, the biological rationale linking radiological features (e.g., irregular borders or increased GGO%) to actual pleural invasion is underexplored.
How could a surgeon or radiologist use DFVP or border type during preoperative planning? Would these features change the surgical approach (lobectomy vs. segmentectomy), or influence adjuvant therapy decisions?
The limitations section, although present, is somewhat superficial. The authors acknowledge the retrospective design and inter-reader variability, but they do not elaborate on the potential influence of CT acquisition heterogeneity, or lack of external validation.
Author Response
Comments 1:
Comments and Suggestions for Authors
This paper investigated the association between CT imaging features and visceral pleural invasion in patients with surgically resected lung adenocarcinoma. It analyzes 643 cases using standard radiologic features and pathology-confirmed visceral pleural invasion.
This a potentially interesting paper. Please see some of the major concerns listed below:
Materials & Methods
Were all CT scans acquired with the same scanner and protocol across the 2011–2015 period? Were parameters like slice thickness, reconstruction algorithm, contrast administration, and radiation dose standardized?
Response 1:
Thank you for your insightful comment. All CT scans in this study were performed at a single institution using multi-detector CT scanners. During the study period, imaging protocols, including slice thickness, reconstruction algorithm, and contrast administration protocols, were standardized (page 4, line 149-153; page 11, line 337-341).
Comments 2:
How were features like “border type” or “tumor shape” operationally defined? Were they visually assessed or quantitatively measured using imaging software?
Response 2:
Thank you for raising this important point. CT imaging features, including tumor shape and border type, were visually assessed by one or two experienced thoracic radiologists through consensus interpretation to ensure consistency and reduce interobserver variability (page 4, line 149-151).
Comments 3:
The dataset is over 10 years old. How do the authors justify the relevance of these findings to current imaging standards, which may now include dual-energy CT or radiomics?
Response 3:
Thank you for your valuable comment. Although the dataset was collected between 2011 and 2015, the fundamental CT imaging characteristics of lung adenocarcinoma and their association with visceral pleural invasion remain clinically relevant. Our study offers important baseline data derived from conventional CT imaging, which continues to be widely utilized in clinical practice worldwide.
Discussion 1:
The authors reference previous studies with similar findings but miss an opportunity to critically contrast their results with conflicting evidence, such as studies questioning the reliability of CT-based VPI prediction for early-stage tumors.
Response 1:
Thank you for your insightful comment. While some previous studies have questioned the reliability of CT-based VPI prediction in early-stage tumors, our findings show that features such as reduced DFVP, irregular tumor borders, and increased GGO proportion are significantly associated with pathological VPI. These results suggest that, with careful interpretation, conventional CT imaging remains a valuable tool for preoperative assessment. (page 10-11, line 283-324).
Discussion 2:
Additionally, the biological rationale linking radiological features (e.g., irregular borders or increased GGO%) to actual pleural invasion is underexplored.
Response 2:
Thank you for your valuable comment. While correlations between radiological features (e.g., irregular borders, increased GGO%) and tumor invasiveness exist, the biological mechanisms linking them to pleural invasion remain unclear. Irregular borders and increased GGO% may reflect tumor growth and invasiveness (page 10, line 229-308). Future research should focus on exploring these underlying molecular mechanisms.
Discussion 3:
How could a surgeon or radiologist use DFVP or border type during preoperative planning? Would these features change the surgical approach (lobectomy vs. segmentectomy), or influence adjuvant therapy decisions?
Response 3:
Thank you for this important question. In clinical practice, radiologists and thoracic surgeons can incorporate imaging features such as DFVP and tumor border type into preoperative planning. Specifically, a tumor exhibiting a small DFVP or an irregular border may raise suspicion for visceral pleural invasion (VPI), prompting the surgical team to favor lobectomy over segmentectomy or wedge resection in order to achieve wider surgical margins and reduce the risk of local recurrence. Moreover, the preoperative identification of features suggestive of VPI could influence the extent of lymph node dissection and guide postoperative management decisions, including the consideration of adjuvant chemotherapy, given that VPI is recognized as an adverse prognostic factor even in early-stage lung adenocarcinoma.
Discussion 4:
The limitations section, although present, is somewhat superficial. The authors acknowledge the retrospective design and inter-reader variability, but they do not elaborate on the potential influence of CT acquisition heterogeneity, or lack of external validation.
Response 4:
Thank you for your valuable feedback. We concur that the limitations section would benefit from further refinement. In addition to the previously noted retrospective design and inter-reader variability, we acknowledge that the heterogeneity in CT acquisition methods may influence the consistency and comparability of imaging data across different patient cohorts. We will revise the limitations section to provide a more comprehensive discussion of these factors and their potential impact on the study's conclusions (page 11, line 333-341).
Round 2
Reviewer 1 Report
Comments and Suggestions for Authors
The corrections of the article by the authors allow for greater clarity in the scientific and research field regarding Predicting visceral pleural invasion in resected lung adenocarcinoma by computed tomography.
However considering the large case history considered, the article with the additional revisions made by the authors acquires a greater quality and excellent suggestions in the clinical management and in the scientific research of this type of disease
Reviewer 3 Report
Comments and Suggestions for Authors
The authors have made the requested adjustments
Reviewer 4 Report
Comments and Suggestions for Authors
Thank you for carefully implementing the required revisions.